# Advancing *Mycobacterium leprae* transmission research: Insights from the R2STOP fund

Marta Sólveig Palmeirim[1,2], Annemieke Geluk[3], Bouke C. de Jong[4], Sofie M. Braet[4], Kevin R. Macaluso[5], JoAnn M. Tufariello[6], Mallika Lavania[7,8], Itu Singh[8], Rahul Sharma[9], Pushpendra Singh[10,11], Peter Steinmann[1,2]*

1 Swiss Tropical and Public Health Institute, Allschwil, Switzerland, 2 University of Basel, Basel, Switzerland, 3 Leiden University Medical Center, Leiden, The Netherlands, 4 Institute of Tropical Medicine, Antwerp, Belgium, 5 University of South Alabama College of Medicine, Mobile, Alabama, United States of America, 6 Department of Microbiology, Icahn School of Medicine at Mount Sinai, New York, New York, United States of America, 7 ICMR-National Institute of Virology, Pune, India, 8 Stanley Browne Research Laboratory, The Leprosy Mission Trust India, New Delhi, India, 9 Advanta Genetics, Tyler, Texas, United States of America, 10 ICMR-National Institute of Research in Tribal Health, Jabalpur, India, 11 Model Rural Health Research Unit, Badoni, Datia, India

* peter.steinmann@swisstph.ch

## Abstract

### Background

Leprosy remains a significant public health burden in many low- and middle-income countries, with the transmission pathways of *Mycobacterium leprae* remaining incompletely understood. The Research to Stop Transmission of Neglected Tropical Diseases (R2STOP) fund was established by two NGOs to address this gap by supporting research projects focused on *M. leprae* transmission. This article outlines R2STOP's selection process for funding projects and summarizes the impact and findings of the resulting research, illustrating the collective progress in understanding *M. leprae* transmission.

### Methodology/Principal findings

The funding priorities established by R2STOP in the call were: (i) human-to-human transmission, (ii) non-human reservoirs, (iii) host-pathogen interactions and (iv) transmission networks. R2STOP allocated a total budget of CAD one million to support research projects focused on *M. leprae* transmission. The selection process involved remote reviews of letters of intent and full proposals, followed by an in-person proposal review meeting where projects were evaluated based on criteria such as significance, innovation, approach, and environmental impact. Final selections were made by a Scientific Review Committee, resulting in the funding of six projects. The funded projects all yielded significant findings from exploring a variety of topics such as persistent transmission in the Comoros islands; the potential role of patients and soil in transmission; ticks' role in transmission to hosts; biomarkers for leprosy

**Data availability statement:** All data is contained in the manuscript itself. Note: this manuscript summarizes research conducted by others so primary data is not available to the authors.

**Funding:** The author(s) received no specific funding for this work.

**Competing interests:** The authors have declared that no competing interests exist

progression; ofloxacin resistance in India; and methods to grow *M. leprae* on axenic media. Twenty-four MSc and PhD students were involved in the six funded research projects, and 29 scientific articles were published.

## Conclusions/Significance

The R2STOP funding scheme played an important role in advancing our understanding of *M. leprae* transmission pathways and showcased the relevance of having funds allocated to this neglected aspect of leprosy control. Relevant research continues to be supported through the Leprosy Research Initiative.

### Author summary

Leprosy is still a health concern in many countries, but how the bacteria that cause the disease are transmitted is not fully understood. We wanted to help fill this gap in knowledge. Through the Research to Stop Transmission of Neglected Tropical Diseases (R2STOP) fund, supported by two non-profit organizations, we provided funding to scientists around the world to study how leprosy spreads. We focused on several key areas, including human-to-human transmission, the role of animals and the environment, and how the bacteria interact with human hosts. After a careful selection process, six research projects were funded. These projects uncovered new insights into how leprosy continues to spread in places like the Comoros islands, possible involvement of soil and ticks, the presence of antibiotic resistance in India, and efforts to grow the bacteria in the laboratory. In addition to scientific discoveries, this program helped train 24 graduate students and resulted in 29 published studies. Our work highlights the importance of targeted funding to improve understanding of leprosy transmission, which is essential for developing better strategies to control and eventually stop the spread of this disease.

## Introduction

Leprosy, caused by a chronic infection with *Mycobacterium leprae*, affects the skin, peripheral nerves, and upper respiratory tract [1]. With a stable number of approximately 200,000 new cases reported each year for over a decade, the disease remains a significant public health burden in several low- and middle-income countries [2]. Control strategies primarily revolve around early detection and appropriate multidrug therapy to cure patients and prevent further transmission [3]. However, a complete understanding of *M. leprae* transmission pathways remains elusive [4]. Although close contact with untreated patients and certain genetic and immune factors play a role, many questions regarding transmission remain unanswered. Key knowledge gaps include the precise modes of transmission, the entry and exit points of the bacteria, the reasons behind the variable incubation period, the apparent host-genetic resistance in certain individuals, and potential environmental

reservoirs or vectors [4,5]. Additionally, socio-economic conditions, such as nutrition, urbanization and overcrowding, may influence disease transmission. To address these gaps, a systematic review was conducted, which identified three modes of transmission: human-to-human via aerosols or direct skin contact (most plausible), direct inoculation through injuries, and transmission from environmental or zoonotic reservoirs and via vectors such as insects [6]. However, none of the reviewed studies unequivocally demonstrated the mechanisms by which *M. leprae* travels from an infectious leprosy patient to another individual.

To help understand *M. leprae*'s transmission pathways, an international symposium titled "*Developing Strategies to Block the Transmission of Leprosy*" was organized in May 2014 by Effect Hope at the National School of Tropical Medicine in Texas. The symposium confirmed the absence of research programs dedicated to transmission, and aimed to establish a research agenda that would bridge the gaps in our understanding of *M. leprae* transmission to complement global efforts for leprosy elimination. A report summarizing the symposium's outcomes was subsequently published [7]. In response to the identified research priorities, Effect Hope partnered with Leprosy Mission Ireland to establish the Research to Stop Transmission of Neglected Tropical Diseases (R2STOP) fund, a funding scheme dedicated to supporting research on *M. leprae* transmission, a field that had arguably been neglected in recent decades.

The current article describes the selection process employed by R2STOP to select research projects to fund. We also present a summary of the findings, impact, and future outlook of the funded projects, demonstrating the collective progress made by this funding scheme concerning the understanding of *M. leprae* transmission.

## Methods

R2STOP allocated a total budget of one million CAD to support research projects focused on *M. leprae* transmission. Grants were awarded within the range of CAD 10,000–300,000, with a maximum duration of three years for each project. The initial funding priorities, which were established based on the literature review [6] and the discussions held in the symposium, were: (i) human-to-human transmission of *M. leprae*, (ii) non-human reservoirs of *M. leprae*, (iii) host-pathogen interactions and (iv) transmission networks.

The call for research proposals was announced on October 23rd, 2015, and interested researchers were invited to submit letters of intent (Fig 1). The remote review of these letters of intent was conducted by seven reviewers who had expertise in the field of leprosy. Upon the selection of letters of intent, the corresponding applicants were invited to submit full

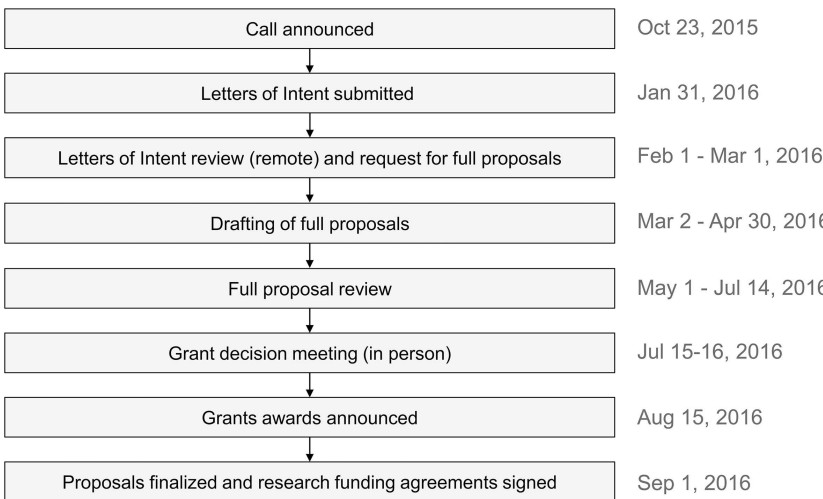

| | |
|---|---|
| Call announced | Oct 23, 2015 |
| Letters of Intent submitted | Jan 31, 2016 |
| Letters of Intent review (remote) and request for full proposals | Feb 1 - Mar 1, 2016 |
| Drafting of full proposals | Mar 2 - Apr 30, 2016 |
| Full proposal review | May 1 - Jul 14, 2016 |
| Grant decision meeting (in person) | Jul 15-16, 2016 |
| Grants awards announced | Aug 15, 2016 |
| Proposals finalized and research funding agreements signed | Sep 1, 2016 |

**Fig 1. Summary of the process followed during the R2STOP research proposal call, review and grant decision.**

proposals by May 1st, 2016. The full proposals underwent an initial remote review, followed by a proposal review meeting held on July 15th-16th, 2016. During the proposal review meeting, the Scientific Review Committee convened to determine which projects would receive funding. Each proposal was evaluated by the primary and secondary reviewers who provided a comprehensive overview of the proposal, along with expert feedback. The reviewers also individually assigned scores to the proposals based on five review criteria: significance, investigators track record and potential, innovation, approach, and environment. Scores were assigned on a scale from one (exceptional) to nine (poor). Additionally, considerations such as safety, ethics, budget, and dissemination plans were taken into account. During the proposal review meeting, the Scientific Review Committee engaged in detailed discussions regarding each proposal. Individual rankings were assigned by each of the seven reviewers, using the same scoring system, resulting in a collective tally of scores to provide a final ranking/grade for each proposal. The grant awards were announced on August 15th, 2016, after which additional information was requested, and proposals were finalized. Research funding agreements were signed on September 1st, 2016 and the grants release began on October 1st, 2016.

## Results

### Funded project characteristics

A total of six projects were awarded funding. Table 1 presents the main characteristics of these projects. Durations of the projects were planned for two years (n = 4) or three years (n = 2). The final funding ranged from CAD

Table 1. Characteristics of the six funded R2STOP projects. SRF, senior research fellow.

| Project title | Principal investigator | Planned duration | Funding awarded (CAD) | Type of study | Funding priorities covered | Nb of publications | Nb of conference contributions | Nb of students involved | Nb of follow-up grants |
|---|---|---|---|---|---|---|---|---|---|
| Improved understanding of ongoing transmission of leprosy in the Comoros, a region hyperendemic for the disease | de Jong | 3 years | 394,890 | Feasibility pre-study | Transmission networks | 3: [8–10] | 5 | 1 | 2 |
| Identification of human susceptibility genes and pathogen-based transmission patterns for human and environmental sources of *M. leprae* in a leprosy endemic area in Bangladesh | Geluk | 3 years | 395,145 | Side study to another larger one | Host-pathogen interactions, transmission networks, non-human reservoirs | 9: [11–19] | 1 | MSc: 0 PhD: 1 | 2 |
| Role of arthropods in transmission of leprosy | Macaluso | 2 years | 176,872 | Stand-alone project | Non-human reservoirs | 1: [20] | 1 | MSc: 0 PhD: 1 | 1 |
| Biomarkers for early detection of leprosy using comparative transcriptomics | Singh | 2 years | 26,000 | Side study to another larger one | Host-pathogen interactions | 1: [21] | 4 | MSc: 5 PhD: 1 SRF: 2 | 1 |
| Genomic markers for pathological variants and transmission of leprosy bacilli using whole genome sequencing | Singh and Sharma | 2 years | 177,776 | Stand-alone project | Transmission networks, host-pathogen interactions | 15: [22–38] | 6 | 11 | 1 |
| *Mycobacterium haemophilum*: a novel system to elucidate *M. leprae*-host Interactions | Tufariello | 2 years | 263,247 | Stand-alone project | Human-to-human transmission, host-pathogen interactions | 0 | 1 | 2 | 0 |

26,000–395,145. The most addressed funding priority was host-pathogen interactions, and the least was human-to-human transmission.

**Funded project key findings**

While each project pursued the common goal of understanding the pathways of *M. leprae* transmission, they encompassed a diverse range of research questions, leading to a wide spectrum of findings.

Identification of human susceptibility genes and pathogen-based transmission patterns for human and environmental sources of *M. leprae* in a leprosy endemic area in Bangladesh (PI: Geluk)**.**

Aim: Identification of host derived biomarkers that prospectively predict those *M. leprae* infected individuals who progress to leprosy to allow early diagnosis, better treatment outcomes and facilitate interventions aimed at stopping bacterial transmission as early diagnosis combined with (prophylactic) treatment will prevent bacterial transmission and life-long disabilities.

Main findings: The study used blood samples of contacts (n = 5,352) collected in previous projects covering 9 years of follow-up in northwest Bangladesh and identified for the first time a prospective transcriptional risk signature in blood predicting development of paucibacillary (PB) leprosy 4–61 months before clinical diagnosis. This biomarker signature, designated RISK4LEP, is composed of 4 genes. Assessment of this signature in contacts of patients can function as an adjunct diagnostic tool to target implementation of interventions to restrain leprosy development. Evaluation of RISK4LEP in other countries and in TB patients is planned to assess global applicability and specificity.

Role of arthropods in transmission of leprosy (PI: Macaluso)**.**

Aim: To assess the ability of *Amblyomma* ticks, which commonly infest both humans and armadillos in the southern United States, to harbor viable *M. leprae* and transmit the pathogen between vertebrate hosts. The association between nine-banded armadillos as a natural reservoir of *M. leprae* and human cases of leprosy suggests the existence of vectors or reservoirs, such as hematophagous arthropods, that help facilitate transmission of *M. leprae* between hosts.

Main findings: Ticks could be infected and sustained vertical (between tick lifecycle stages) transmission was observed as well as transmission by feeding ticks to the vertebrate host. Additionally, viable culture of *M. leprae* for up to 49 days using a tick-derived cell line could be demonstrated [20]. Transcription analyses could not be conducted. With colleagues at the National Hansen's Disease Program, this research is pursued with further elucidation of tick infection and transmission.

Improved understanding of ongoing transmission of leprosy in the Comoros, a region hyperendemic for the disease (PI: de Jong)**.**

Aim: The primary aim of the project was to identify individuals in Anjouan and Moheli, Comoros, who would have derived the greatest benefit from prophylactic treatment. As part of the study, the program reintroduced sampling of leprosy patients for laboratory confirmation using molecular techniques.

Main findings: Proximity to an index case in the last 5 years was found to be a significant risk factor for incident leprosy, in a dose response relationship [39]. Ongoing molecular epidemiological analyses of *M. leprae* genotypes identified clustering at village level, and that different genotypes are circulating within one village.

The high diagnostic confirmation rate (67.5% of PB and 80.1% of MB patients) confirmed the accuracy of the clinical leprosy diagnosis. Comparing various types of samples, less invasive samples like nasal swabs and finger prick blood, were found to correlate well with the higher bacterial loads determined in skin biopsies [8].

Using the same biopsies, we validated a molecular genotyping technique, Deeplex Myc-Lep [10]. With Deeplex Myc-Lep analyses on the biopsies, the first-ever drug resistance survey was conducted, indicating that drug-resistant *M. leprae* is absent in the Comoros, at least with regard to currently known genetic markers [40]. This excludes treatment failure as cause for persistent hyperendemicity [9].

Biomarkers for early detection of leprosy using comparative transcriptomics (PI: Singh)**.**

Aim: To investigate the transcriptional changes in the peripheral blood mononuclear cells (PBMCs) associated with early stages of leprosy progression in experimentally infected armadillos to enable the development of diagnostic

tests for leprosy. Most armadillos develop disseminated disease within 2 years of experimental infection; however, it has been reported that roughly 20% of animals can resist even high-dose experimental inoculation. We explored this differential susceptibility of armadillos to compare the gene expression pattern of the leprosy-resistant vs -susceptible animals to identify characteristic gene expression patterns associated with disease progression in the susceptible host.

Main outcomes: Differentially expressed genes (DEGs) between the resistant and susceptible armadillos were identified. DEGs related to neurological and immunological terms identified IDO-1, mTOR signalling, TLR, T-cell, MAPK and Notch Pathways as important players in leprosy progression in susceptible hosts. Such genes/pathways can be candidate gene signatures of leprosy progression. Further validation of such biomarkers in patient samples can be useful.

Genomic markers for pathological variants and transmission of leprosy bacilli using whole genome sequencing (PI: Singh and Sharma).

Aim: To develop enrichment assays that offer sufficient coverage of the *M. leprae* genome directly from clinical specimens of paucibacillary, as well as multibacillary leprosy patients. The conventional PCR-sequencing methods are usually unable to differentiate various *M. leprae* strains. Therefore, genome-wide analysis is important. However, the host-derived tissue material (either human biopsies or mouse foot pad/armadillo-derived tissues) invariably possesses vast amounts of host DNA, which means enrichment methods are required to allow sufficient coverage of *M. leprae* strains using next-generation sequencing (NGS) methods.

Main results: Methods for the depletion of host tissues were adapted, followed by sequencing. In addition, PCR-amplifiable biotinylated baits for hybridization-based capture of the target DNA were developed. These approaches allowed genome coverage of *M. leprae* strains. The genome-wide comparison of these strains has revealed novel genomic markers for improved molecular epidemiological assays. This will help in identifying the genomic markers suitable to monitor local transmission of infection and pathological variations of leprosy bacilli.

*Mycobacterium haemophilum*: a novel system to elucidate *M. leprae*-host interactions (PI: Tufariello).

Aim: To expand upon a previous observation that *M. haemophilum* is the closest culturable relative of *M. leprae* among sequenced mycobacterial species. The inability to propagate *M. leprae* in vitro on axenic media represents a significant impediment to the genetic manipulation and study of the organism. *M. haemophilum* shares with *M. leprae* a number of important features, including deficits in iron acquisition pathways, which may limit the ability of the organisms to grow on standard culture media.

Main outcomes: Vectors were created and tested for delivering genes into *M. leprae* that might identify essential growth requirements. Much of the work was tested on *M. haemophilum*. Successes have included transfer of genes to *M. haemophilum* from another mycobacterium that allowed for its growth on an artificial medium that otherwise did not sustain growth, supporting the concept that this approach has potential for *M. leprae*. Focusing on *M. haemophilum* as a surrogate, fluorescent reporter genes were introduced, the bacterium was transduced with mycobacteriophages and a Schwann cell infection model was established.

Textbox 1 outlines the principal investigators' key takeaway messages, providing a comprehensive summary of the results of the six funded R2STOP projects.

## Principal investigators' perceptions of the R2STOP funding scheme

R2STOP consisted of a single call, no second call took place. However, the priority of research on the transmission of leprosy, which had originally been included in the Leprosy Research Initiative (LRI) focus areas and was put on hold during R2STOP, was then reinstated by LRI.

Concerning specific aspects of the process, all respondents in response to a questionnaire reported being satisfied (either "very satisfied" or "somewhat satisfied") with the application phase and regular reporting/administration.

**Textbox 1. Added insights/knowledge' reported by each of the principal investigators of each of the six funded R2STOP projects.**

| | |
|---|---|
| de Jong | The COMLEP study in the Comoros revealed persistent *M. leprae* transmission, despite effective treatment, emphasizing the complexity of the disease dynamics. The study validated diagnostic techniques and laid the groundwork for subsequent research on post exposure prophylaxis and transmission patterns. |
| Geluk | Using cutting-edge technology, the study found *M. leprae*-DNA in contacts of leprosy patients and soil, indicating that both sources may play a role in transmission. Also, a new *M. leprae* genotype, unique for Bangladesh, was detected, and the first host transcriptomic signature in blood predicting PB leprosy was identified. |
| Macaluso | Ticks could be infected and sustained vertical (between tick lifecycle stages) transmission was seen as well as transmission by feeding ticks to the vertebrate host. Additionally, viable culture of *M. leprae* for up to 49 days using a tick-derived cell line was demonstrated. |
| Singh | Candidate biomarkers associated with leprosy progression were identified using comparative analysis of transcriptional changes in leprosy-susceptible and -resistant armadillos (after experimental infection with leprosy bacilli), paving the way to developing tools for early detection of leprosy. |
| Singh and Sharma | In addition to genotype 1D, which is the predominant genotype in India, the presence of other genotypes like 1B, 1C was also observed. Ofloxacin resistance was detected in new cases in Purulia, West Bengal, India which is a threat to effective second line treatment for rifampicin resistant leprosy cases. For the first time, *M. lepromatosis* was isolated from erythema nodosum leprosum (ENL) cases in India. |
| Tufariello | To overcome the limitation that *M. leprae* cannot be grown in axenic culture, hampering the study of its pathogenesis, *M. haemophilum*'s (the most closely related mycobacterium that can be grown *in vitro*) complete genome was sequenced, and the capacity to introduce fluorescent reporter genes and transduce the bacterium with mycobacteriophages was demonstrated. A Schwann cell infection model was also established. |

Regarding exposure/networking and the possibility to do something that otherwise would be difficult to get funded, all respondents reported to be satisfied, except for one that, in both cases, reported to be "neither satisfied nor dissatisfied".

When asked about their advice for a similar funding scheme, principal investigators mentioned: having four-year support for PhD students within a larger research project, which would simultaneously allow a young researcher to get acquainted with leprosy and obtain a PhD, and having more regular calls, which would allow researchers to plan well thought out proposals.

## Discussion

The R2STOP fund was created in response to the substantial gap in our understanding of *M. leprae* transmission which is an important impediment to the current efforts to control leprosy and interrupt *M. leprae* transmission [41]. The R2STOP funding scheme has played an important role in addressing some of these knowledge gaps by supporting six projects that collectively explored a diverse range of research questions concerning *M. leprae* transmission. These six funded projects addressed different aspects linked to transmission, of varying technical complexity and under a range of conditions. They contributed significantly to our understanding of *M. leprae* transmission pathways, from showing that despite effective treatment in the Comoros, there is persistent *M. leprae* transmission; to finding that perhaps both leprosy patient and soil could play a role in transmission; to investigating the role that ticks may play in sustaining transmission to vertebrate hosts; to identifying candidate biomarkers associated with leprosy progression; to detecting ofloxacin resistance in West Bengal, India, posing a challenge to the efficacy of second-line treatment for leprosy cases resistant to rifampicin, and establishing methods to introduce genes into related organisms in an effort to understand how *M. leprae* may one day be grown on axenic media. Projects varied considerably with regard to funding level and technical challenges, and the projects by Singh suffered from delays due to staff and host institution changes. They were also gravely impacted by effects of the Covid-19 pandemic. For further critical breakthroughs in axenic culture and tick-linked transmission models, funding for follow-up studies was unavailable.

The collaborative effort between Effect Hope and The Mission to End Leprosy has proven instrumental in fostering research activities dedicated to *M. leprae* transmission and succeeded in mobilizing additional funding. Expanding on the

outcomes of the COMLEP (Comoros Leprosy) study and the utilization of diagnostic sampling along with a mobile questionnaire tool (which also incorporated GPS coordinates), subsequent research proposals were supported by the European & Developing Countries Clinical Trials Partnership (EDCTP), on different approaches to providing Post Exposure Prophylaxis (PEP) in the PEOPLE study (clinical trials.gov/NCT03662022) and the BE-PEOPLE study (clinical trials.gov/NCT05597280), allowing the team to further investigate leprosy and its control in the Comoros. The R2STOP partnership has thus paved the way towards the ongoing allocation of resources and expertise to investigating *M. leprae* transmission. Also, LRI has now re-integrated the topic of "transmission" in its annual call for proposals. Indeed, the 2023 call had a specific focus on transmission-related research. Also, in the frame of the LRI Spring meeting 2023, a half-day thematic session was dedicated to revisiting the current understanding of *M. leprae* transmission and reviewing priority funding needs in the light of the conclusions of the meeting in 2014. A particular strength of the R2STOP fund was the ongoing involvement of the scientific review committee which brought together renowned experts in the field. Its members reviewed the annual progress reports, and feedback was provided to the grantees. Also, the PIs or their representatives were invited to attend the LRI spring meeting, fostering dissemination and collaboration with other leprosy-focused research projects. A limitation was the inability to offer follow-up grants even for the most promising projects, and delays along the implementation process that significantly extended the conclusion of the projects beyond the originally projected end of the funding scheme.

In conclusion, the R2STOP fund succeeded in invigorating a long-neglected field of research. To do so, it established a solid infrastructure for the review of submissions and project progress, ensured by recognized experts in the field. Supported by relatively small organizations with a mission to support international health cooperation in the field of leprosy control, the effort kick-started research in a long-neglected field. Integration into LRI with its established infrastructure and mechanisms was a logical next step and ensures long-term sustainability of the research focus.

## Acknowledgments

The authors would like to thank the Scientific Review Committee of R2STOP – Tom Gillis (past Chair), Peter Steinmann (current Chair), Paul Saunderson, Erwin Schuur, Graham Medley, Steve Ault, Abraham Aseffa, Eliane Ignotti - and administrative staff of Effect Hope who contributed to the selection process and management of the R2STOP research projects.

R2STOP thanks the Principal Investigators/ research award grantees, JoAnn Tufariello, Kevin Macaluso, Annemiek Geluk, Bouke de Jong, Pushpendra Singh, Itu Singh, and Mallika Lavania for conducting the research studies.

R2STOP expresses sincere appreciation to the leprosy patients and their contacts whose participation was invaluable to the studies.

## Author contributions

**Conceptualization:** Peter Steinmann.

**Data curation:** Marta Sólveig Palmeirim, Annemieke Geluk, Bouke C. de Jong, Sofie M. Braet, Kevin R. Macaluso, JoAnn M. Tufariello, Mallika Lavania, Itu Singh, Rahul Sharma, Pushpendra Singh, Peter Steinmann.

**Formal analysis:** Marta Sólveig Palmeirim, Peter Steinmann.

**Methodology:** Peter Steinmann.

**Project administration:** Marta Sólveig Palmeirim, Peter Steinmann.

**Supervision:** Peter Steinmann.

**Validation:** Annemieke Geluk, Bouke C. de Jong, Sofie M. Braet, Kevin R. Macaluso, JoAnn M. Tufariello, Mallika Lavania, Itu Singh, Rahul Sharma, Pushpendra Singh, Peter Steinmann.

**Visualization:** Marta Sólveig Palmeirim.

**Writing – original draft:** Marta Sólveig Palmeirim, Annemieke Geluk, Bouke C. de Jong, Sofie M. Braet, Kevin R. Macaluso, JoAnn M. Tufariello, Mallika Lavania, Itu Singh, Rahul Sharma, Pushpendra Singh, Peter Steinmann.

**Writing – review & editing:** Marta Sólveig Palmeirim, Peter Steinmann.

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
