## [Decision Letter · Decision Letter 0]

6 May 2025

Advancing *Mycobacterium leprae* transmission research: insights from the R2STOP fund

Dear Dr. Steinmann,

Thank you for submitting your manuscript to PLOS Neglected Tropical Diseases. After careful consideration, we feel that it has merit but does not fully meet PLOS Neglected Tropical Diseases's publication criteria as it currently stands. Therefore, we invite you to submit a revised version of the manuscript that addresses the points raised during the review process.

Please submit your revised manuscript within 60 days Jul 05 2025 11:59PM. If you will need more time than this to complete your revisions, please reply to this message or contact the journal office at plosntds@plos.org. Please include the following items when submitting your revised manuscript:

We look forward to receiving your revised manuscript.

Kind regards,

Joseph M. Vinetz

Section Editor

Joseph Vinetz

Section Editor

Shaden Kamhawi

co-Editor-in-Chief

Paul Brindley

co-Editor-in-Chief

**Journal Requirements:**

At this stage, the following Authors/Authors require contributions: Marta Sólveig Palmeirim, Annemieke Geluk, Bouke C. de Jong, Sofie M. Braet, Kevin R. Macaluso, JoAnn M. Tufariello, Mallika Lavania, Itu Singh, Rahul Sharma, Pushpendra Singh, and Peter Steinmann. Please ensure that the full contributions of each author are acknowledged in the "Add/Edit/Remove Authors" section of our submission form.

2) Please amend your detailed Financial Disclosure statement. This is published with the article. It must therefore be completed in full sentences and contain the exact wording you wish to be published. Please ensure that the funders and grant numbers match between the Financial Disclosure field and the Funding Information tab in your submission form. Note that the funders must be provided in the same order in both places as well.

**Reviewers' Comments:**

Reviewer's Responses to Questions

**Key Review Criteria Required for Acceptance?**

**Methods:**

-Are the objectives of the study clearly articulated with a clear testable hypothesis stated?

-Is the study design appropriate to address the stated objectives?

-Is the population clearly described and appropriate for the hypothesis being tested?

-Is the sample size sufficient to ensure adequate power to address the hypothesis being tested?

-Were correct statistical analysis used to support conclusions?

-Are there concerns about ethical or regulatory requirements being met?

Reviewer #1: The introduction and methods sections are clearly written and appropriate. I have no further comments on these parts.

**Results**

-Does the analysis presented match the analysis plan?

-Are the results clearly and completely presented?

-Are the figures (Tables, Images) of sufficient quality for clarity?

Reviewer #1: Regarding the results section, I have the following comments and suggestions:

• The amount of funding for the 6 projects varies considerably, ranging from CAD 26,000 to 395,145. However, to interpret the outcomes of each project, it would be important to know how much funding each project received individually. This could be added to Table 1.

• Under the heading ‘Funded project key findings’, the information on each project could be structured more consistently. Now the type and amount of information per project differs considerably, giving the impression of simple copy-paste from other texts. I would suggest 3 standard headings for each description:

1. Title of the project (with name PI)

2. Aim(s) of the study

3. Main findings

With this description I would also suggest that a word count is given to the writers, so that each description is about the same length. In this suggested brief abstract, there is no mention of background, methods, and discussion. The background is usually clear from the title. If necessary, the methodology can be indicated briefly in the aim. (E.g. the aim of our study is to …… by means of ….). Regarding discussion, see my next points.

• I think the box with key messages is too diversely formulated for an easy comparison and overview. I like the idea of the boxes and suggest rephrasing it from ‘key messages’ (which I think is vague) to ‘added insight/knowledge’. This is like what journals sometimes ask authors to include in their summary information: what is known and what is the added knowledge generated by this study. The part on ‘what is known’ can be left out here. But the added insight/knowledge is key to the aim of this report and of most interest to the readers. This could be summarized in the boxes with 2-3 bullet points per project.

• After this review of contents and added knowledge, the second part of the results could then focus on operational aspects of the R2STOP funding scheme (PI’s perception). Given the limited amount of meaningful information, I would shorten this part by leaving out Figure 2 and just give a written description without making it sound like a formal questionnaire study (“all respondents …”). I guess the ‘respondents’ are actually the co-authors, so just keep the text straightforward regarding satisfaction on the four aspects mentioned.

**Conclusions**

-Are the conclusions supported by the data presented?

-Are the limitations of analysis clearly described?

-Do the authors discuss how these data can be helpful to advance our understanding of the topic under study?

-Is public health relevance addressed?

Reviewer #1: Regarding the Discussion and Conclusions, I have the following comments:

• I think the discussion is the part where more critical appraisal is possible. Now there are phrases as “we believe that…”, and “the 6 funded projects contributed significantly to our understanding… (says who…!?)”. These and other phrases in the first paragraph of the discussion are just too general and subjective. If the PIs state their results clearly and concretely in textbox 1, it becomes easier to refer to this ‘added knowledge’ and conclude creditably about their contribution.

• I also suspect that not all projects went equally well. As a reader, I am (for instance) wondering about the projects of Macaluso and Singh. They both report only 1 publication, while both are 2-year projects. Yet Macaluso has 1 PhD student, and Singh has 8 students (of different levels) involved. So, what went on? It might be explained by differences in budget, or many other things. I would like to know more to understand this better. If there are poor or disappointing results, this could be mentioned (tactfully), because there could be important learning points.

• What are ‘strengths and weaknesses’ in the scheme that could be highlighted? I am sure there are things to be said about budget size, procedure, collaboration, capacity building, risks, etc. What are lessons learnt? Do’s and don’ts, etc.? There is some discussion on this in the second paragraph of the Discussion, but I believe it can be more systematic and elaborate.

• The conclusion introduces several elements that were not really mentioned or reviewed in the previous parts, such as ‘invigorating research’, and ‘establishing solid review infrastructure’. I am sure this is the case, but it was not really addressed and established explicitly in the Results and Discussion sections.

**Editorial and Data Presentation Modifications?**

Reviewer #1: The report on page 12 of Sharma and Singh appears to have an incomplete sentence (line 10-12). Also line 6 should read strains and not strain. However, if the authors rearrange these descriptions according to my suggestions, such unclear sentences would hopefully be solved.

**Summary and General Comments**

Reviewer #1: In summary, I believe this report can be greatly improved by introducing a more rigorous structure, especially in the results and discussion sections. Being a self-assessment, a more precise report on the projects and a more detailed critical appraisal of the outcomes and impact of the R2STOP fund would benefit information provision and credibility enormously.

PLOS authors have the option to publish the peer review history of their article (what does this mean? ). If published, this will include your full peer review and any attached files.

**Do you want your identity to be public for this peer review?** For information about this choice, including consent withdrawal, please see our Privacy Policy .

Reviewer #1: **Yes: ** Jan Hendrik Richardus

**Figure resubmission:**

**Reproducibility:**



---

## [Decision Letter · Decision Letter 1]

17 Jul 2025

Dear Dr. Steinmann,

We are pleased to inform you that your manuscript 'Advancing *Mycobacterium leprae* transmission research: insights from the R2STOP fund' has been provisionally accepted for publication in PLOS Neglected Tropical Diseases.

Best regards,

Joseph M. Vinetz

Section Editor

Joseph Vinetz

Section Editor

Shaden Kamhawi

co-Editor-in-Chief

Paul Brindley

co-Editor-in-Chief

Reviewer's Responses to Questions

**Key Review Criteria Required for Acceptance?**

**Methods**

-Are the objectives of the study clearly articulated with a clear testable hypothesis stated?

-Is the study design appropriate to address the stated objectives?

-Is the population clearly described and appropriate for the hypothesis being tested?

-Is the sample size sufficient to ensure adequate power to address the hypothesis being tested?

-Were correct statistical analysis used to support conclusions?

-Are there concerns about ethical or regulatory requirements being met?

Reviewer #1: (No Response)

**Results**

-Does the analysis presented match the analysis plan?

-Are the results clearly and completely presented?

-Are the figures (Tables, Images) of sufficient quality for clarity?

Reviewer #1: (No Response)

**Conclusions**

-Are the conclusions supported by the data presented?

-Are the limitations of analysis clearly described?

-Do the authors discuss how these data can be helpful to advance our understanding of the topic under study?

-Is public health relevance addressed?

Reviewer #1: (No Response)

**Editorial and Data Presentation Modifications?**

Reviewer #1: (No Response)

**Summary and General Comments**

Reviewer #1: The authors have revised the manuscript carefully and systematically according to my comments. I think it now represents a strong, meaningful and interesting report of the R2STOP funding scheme.

PLOS authors have the option to publish the peer review history of their article (what does this mean? ). If published, this will include your full peer review and any attached files.

**Do you want your identity to be public for this peer review?** For information about this choice, including consent withdrawal, please see our Privacy Policy .

Reviewer #1: **Yes: ** Jan Hendrik Richardus

---

## [Editor Report · Acceptance letter]

Dear Dr. Steinmann,

We are delighted to inform you that your manuscript, "Advancing *Mycobacterium leprae* transmission research: insights from the R2STOP fund," has been formally accepted for publication in PLOS Neglected Tropical Diseases.

Best regards,

Shaden Kamhawi

co-Editor-in-Chief

Paul Brindley

co-Editor-in-Chief
